# Protein Disulfide Isomerase A3 Regulates Influenza Neuraminidase Activity and Influenza Burden in the Lung

**DOI:** 10.3390/ijms23031078

**Published:** 2022-01-19

**Authors:** Nicolas Chamberlain, Mona Ruban, Zoe F. Mark, Sierra R. Bruno, Amit Kumar, Ravishankar Chandrasekaran, Dhemerson Souza De Lima, Danielle Antos, Emily M. Nakada, John F. Alcorn, Vikas Anathy

**Affiliations:** 1Department of Pathology and Laboratory Medicine, Larner College of Medicine, The University of Vermont, Burlington, VT 05405, USA; nicolas.chamberlain@yale.edu (N.C.); mona.ruban@uvm.edu (M.R.); zoe.mark@med.uvm.edu (Z.F.M.); Sierra.Bruno@med.uvm.edu (S.R.B.); amit.kumar@med.uvm.edu (A.K.); dhemerson.souza-de-lima@med.uvm.edu (D.S.D.L.); emily.nakada@mail.mcgill.ca (E.M.N.); 2Department of Medicine, Larner College of Medicine, The University of Vermont, Burlington, VT 05405, USA; ravishankar.chandrasekaran@med.uvm.edu; 3Department of Pediatrics, Division of Pulmonary Medicine, UPMC Children’s Hospital of Pittsburgh, University of Pittsburgh, Pittsburgh, PA 15260, USA; d.antos@pitt.edu (D.A.); john.alcorn@chp.edu (J.F.A.)

**Keywords:** influenza, PDIA3, NA, disulfides, LOC14

## Abstract

Influenza (IAV) neuraminidase (NA) is a glycoprotein required for the viral exit from the cell. NA requires disulfide bonds for proper function. We have recently demonstrated that protein disulfide isomerase (PDI)A3 is required for oxidative folding of IAV hemagglutinin (HA), and viral propagation. However, it not known whether PDIs are required for NA maturation or if these interactions represent a putative target for the treatment of influenza infection. We sought to determine whether PDIA3 is required for disulfide bonds of NA, its activity, and propagation of the virus. Requirement of disulfides for NA oligomerization and activity were determined using biotin switch and redox assays in WT and PDIA3^−/−^ in A549 cells. A PDI specific inhibitor (LOC14) was utilized to determine the requirement of PDIs in NA activity, IAV burden, and inflammatory response in A549 and primary mouse tracheal epithelial cells. Mice were treated with the inhibitor LOC14 and subsequently examined for IAV burden, NA activity, cytokine, and immune response. IAV-NA interacts with PDIA3 and this interaction is required for NA activity. PDIA3 ablation or inhibition decreased NA activity, viral burden, and inflammatory response in lung epithelial cells. LOC14 treatment significantly attenuated the influenza-induced inflammatory response in mice including the overall viral burden. These results provide evidence for PDIA3 inhibition suppressing NA activity, potentially providing a novel platform for host-targeted antiviral therapies.

## 1. Introduction

The influenza A virus (IAV) causes severe respiratory illness and worldwide impact [1]. While vaccination and therapeutics are available, they are often rendered ineffective due to the accumulation of mutations in the viral genome [1]. One potential strategy to circumvent this type of mutational resistance is targeting the host proteins or post-translational processes utilized by the virus during propagation. Literature on the interaction between IAV and host protein disulfide isomerases (PDIs), involved in disulfide bond formation, is emerging, and viruses often utilize redox-active host chaperone proteins to assist in their protein folding [2,3,4,5,6,7].

PDIs are a major disulfide catalyzing family of enzymes in mammalian cells [8]. One isoform, PDIA3, is known to be required for efficient folding of IAV hemagglutinin (HA) in vitro [2]. We have recently demonstrated PDIA3 is required for catalyzing disulfide bonds in HA and active infection in vivo [3]. A recent study has also shown that specific overexpression of PDIA3, rather than other PDI isoforms, yields improved stability of HA [9]. While other PDI isoforms are known to play a role in IAV protein expression [4], the exact nature of that role remains unknown. 

We have shown that treatment with the reversible PDI inhibitor LOC14 [10] in vitro significantly alters the oxidative folding of IAV HA, leading to disruptions in HA maturation and virus-induced inflammatory cytokine production [3]. However, whether these disruptions in disulfide bond formation are limited to HA remain unknown. To further gain insight, we investigated the effect of PDIA3 on IAV-NA. NA is a homotetrameric enzyme found on the surface of the virion [11]. NA is a glycoprotein stabilized by 32 disulfide bonds once fully matured and is known to traffic through the endoplasmic reticulum (ER), the primary site of localization of PDI [8,12]. 

Active NA is required for viral release, cleaving sialic acid residues from cell surface proteins and lipids to prevent viral aggregation on the cell [11,12]. NA inhibitors are currently front-line therapeutic options for influenza infections [12]; however, their effectiveness is often limited by rapid viral mutation [13] and a limited window of application. 

In this study, we provide evidence that PDIA3 and IAV-NA interact. Inhibition or ablation of PDIA3 results in a reduction of NA disulfide bonds, oligomerization, and activity. Treatment of IAV-infected mice or lung epithelial cells with LOC14 decreased viral burden, inflammation, and pro-inflammatory cytokine production. These results demonstrate that PDIA3 regulates IAV-NA activity, and inhibiting PDIs in vivo decreases both IAV burden and subsequent pro-inflammatory responses.

## 2. Results

### 2.1. Disulfide Bonds Are Required for NA Activity, and PDIA3 Interacts with IAV NA

Airway epithelial cells are the primary site of infection and replication of influenza [14], therefore we utilized airway or lung epithelial cells in our in vitro experiments. Influenza-NA is responsible for cleaving sialic acid residues of glycoproteins, enabling viral exit from the infected cells [11,12], disulfide bonds stabilize the structure and are required for the proper function of NA [12,15]. Comparisons of cysteine residues and disulfide bonds show remarkable conservation in various IAV strains, and recombinant-NA (rNA) activity was decreased by the reducing agent DTT (Figure 1A,B). Co-immunoprecipitation experiments after IAV infection or transfection with NA expressing plasmids revealed that PDIA1 and PDIA3 interact with NA compared to other PDIs in the IAV infection model, and PDIA3 interacted with the ectopically expressed NA (Figure 1C–E). To determine whether PDIA3 expression affects NA disulfide bonds and activity, PDIA3 ablated A549 cells were infected with either H1N1-PR8 or transfected with NA (N1)-expressing plasmid. We found significant decreases in NA production (~20%) (Figure 1F,G). A biotin-based cysteine sulfhydryl labeling methodology [3] (Figure 1H) revealed that disulfide bonds of NA were decreased in *PDIA3*^−/−^ compared to WT cells (Figure 1I,J). The production of IFNβ was also decreased in *PDIA3*^−/−^ compared to WT cells (Figure 1K). These results indicated that PDIA3 and NA interact, and disulfide bond formation in NA depends on PDIA3. Deleting PDIA3 also decreased IFNβ response potentially associated with alterations in viral burden. 

### 2.2. LOC14 Treatment Alters Disulfides and Activity of NA and Decreases the Viral Burden

LOC14 inhibits both PDIA1 and A3 [10,15]. We observed that NA binds to both PDIA1 and A3. To test the impact of PDIA1 or A3 inhibitor on NA, lung epithelial cells were infected with IAV and treated with LOC14 (Figure 2A). Measurement of dead cell protease in the culture supernatants showed that LOC14 significantly decreased cell death in mock or influenza-infected A549 cells compared to DMSO-treated samples (Figure 2B). We observed a significant decrease in the production, disulfide bonds, and activity of NA in A549 cells (Figure 2B–D). Analyses of samples from primary mouse tracheal epithelial cells (MTECs) infected with IAV and treated with LOC14 revealed a decrease in oligomerization and activity of NA along with a decrease in IAV burden as quantitated by the levels of IAV-PA mRNA (Figure 2E–G). Interestingly, we also observed decreased pro-inflammatory cytokine production, including IFNβ-associated genes (Irf7), CCL20, and IFNβ in LOC14-treated cells compared to VC-treated IAV-infected cells (Figure 2H,I). These results indicate that a decrease in the formation of disulfide bonds of NA leads to impaired NA oligomerization, decreased activity, IAV burden, and subsequent decreases in pro-inflammatory responses in IAV-infected lung epithelial cells.

### 2.3. LOC14 Decreases IAV Burden and Inflammation in Mice

C57BL/6NJ mice were treated with LOC14 and infected with H1N1 PR8 (Figure 3A). Analysis of inflammatory cells and cytokines in BALF showed a significant decrease in inflammatory cell profiles in LOC14-treated mice infected with IAV compared to DMSO-mice infected with IAV, including IFN-associated Irf7 transcript in lung homogenates (Figure 3B–F). Cytokine analysis in BALF samples showed a significant decrease in cytokine profiles in LOC14-treated mice infected with IAV compared to DMSO-mice infected with IAV (Figure 3G–J). Furthermore, compared to all the other groups, we observed significant elevation of BALF protein levels in IAV-infected DMSO-treated mice. A higher concentration of LOC14 (50 mg/kg) treatment significantly attenuated IAV-induced BALF protein (Figure 3K). There were no significant alterations in the serum alanine transaminase (ALT) levels (Figure 3L). These results indicated that LOC14 did not induce epithelial barrier disruption or systemic toxicity compared to vehicle-treated mice. 

Western blot and RT-qPCR analysis of viral protein (HA), transcripts (PA) in isolated lungs as well as re-infection and plaque assays with lung homogenates, showed significantly decreased viral burden in LOC14-treated mice compared to untreated mice (Figure 4A–E). Analysis of NA activity in the BALF of infected mice indicated a significant decrease in the higher LOC14-treated group (Figure 4F). Taken together, these results indicated that LOC14 treatment decreases viral burden, NA activity, and subsequent inflammatory response in vivo. 

## 3. Discussion

Our findings here show that the glycoprotein-specific PDIA3 interacts with NA, and ablation of PDIA3 in lung epithelial cells results in decreases in disulfide bonds, production, and activity of NA, subsequently, resulting in decreased viral burden and pro-inflammatory response during influenza infection. Treatment with LOC14, a reversible inhibitor of PDIA1/A3 [16], alters the oxidative folding and oligomerization of NA resulting in loss of activity. LOC14 treatment of IAV-infected mice significantly decreased both viral load and subsequent inflammatory response. Collectively, these results suggest the importance of PDIA3 in the oxidative folding of NA and demonstrate the impact of host PDIA3 in viral maturation and subsequent inflammation. 

The influenza virus causes seasonal outbreaks and occasional pandemics resulting in significant morbidity and mortality [16,17,18,19]. Besides low efficacy of vaccination, there are only a few FDA approved influenza specific anti-viral drugs that are available for treatment [20,21,22]. Influenza strains are showing resistance to these available drugs [20,21,22,23,24,25,26,27,28]. Here we suggest a potential strategy to circumvent development of resistance to existing drugs or vaccine shortcomings is to target the host proteins or post-translational processes utilized by the virus during propagation. 

PDIA3 catalyzes oxidative modifications of numerous proteins, including proteins that are mediators of respiratory diseases [29,30,31,32,33]. PDIA3 ablation or inhibition resulted in decreases in allergic airway inflammation, airway hyperresponsiveness, peribronchiolar fibrosis, and pulmonary fibrosis. These phenotype improvements were attributed to deficiencies in growth factors (POSTN, EGF, and SPP1), chemoattractant (Eotaxin), and decreases in death receptor, Fas activity [30,31,32,33]. We also observed that PDIA3 depletion decreases influenza-induced apoptosis of lung epithelial cells and disulfide bonds in HA, leading to decrease in influenza burden and airway hyperresponsiveness [3,30]. Here we report for the first time that PDIA3 inhibition disrupts disulfide bonds in influenza-NA, and ultimately its activity. This mode of inhibition of NA subsequently decreases influenza burden in cells and mouse lungs. 

Another important component of our work is a dose-dependent decrease in many cytokines/and chemokines in the BAL fluid of LOC14-treated mice. Cytokines and chemokines are often stabilized by disulfide bonds [33,34,35,36], and these bonds have been shown to be important for their function [33,34,35,36]. Although PDIs direct role in the immunomodulatory effect needs a thorough examination, given that PDIs are a major disulfide catalyzing enzymes in the cell [8], it is likely to play a major role in immunomodulation by catalyzing the disulfide bond formation in cytokines, chemokines, and growth factors such as osteopontin [31,32]. Treatment with NA inhibitors can also yield similar decreases in immune activation [37], this is likely due to lower viral levels, as NA inhibitors are not known to directly modulate the immune system [38] and their limited effectiveness when used after the period of initial viral replication [37,38,39,40]. Our studies do not show long-term effects of LOC14 treatment in influenza-induced immune response and lung function. Next, it is not clear whether the decrease in inflammatory response by PDIA3 inhibition is due to attenuation of virus or a direct effect on inflammatory molecules. Furthermore, future studies are required to ascertain whether using PDIA3 inhibitors is a better strategy during IAV infection compared to NA inhibitors or there is a synergistic effect by using both. Despite the above-mentioned limitations, our novel observation in this study would provide insight into a host-based strategy to treat IAV-infection and inflammation in the future. 

## 4. Material and Methods

### 4.1. Ethics Statement

All animal studies were approved by the University of Vermont Institutional Animal Care and Use Committee and carried out in accordance with the Guide for the Care and Use of Animals of the National Institutes of Health [3]. The University of Vermont adheres to the “U.S. Government Principles for the Utilization and Care of Vertebrate Animals Used in Testing, Research, and Training”, “PHS Policy on Humane Care and Use of Laboratory Animals”, “USDA: Animal Welfare Act & Regulations”, and “the Guide for the Care and Use of Laboratory Animals”. The University is accredited by the Association for Assessment and Accreditation of Laboratory Animal Care International (AAALAC). University of Vermont’s PHS Assurance Number: A3301-01, expiration date: 31 October 2021. University of Vermont IACUC was approved on 12 October 2018, and the animal protocol number is PROTO202000102, IACUC Legacy number 19-005 [3]. The de novo renewal was approved on 4 June 2020, and the animal protocol number is PROTO202000102, IACUC Legacy number 19-005 [3].

### 4.2. Viruses

Influenza A virus Puerto Rico 8/34 (H1N1) (10100374) was purchased from Charles River (Wilmington, MA, USA) [3].

### 4.3. Cells and Treatments

Primary MTECs were isolated and cultured from age and sex-matched wild type (WT) C57BL/6NJ mice as previously described [3]. Cells were plated at 2 × 10^6^ cells/dish and when greater than 90% confluent, infected with mouse-adapted H1N1 influenza A virus Puerto Rico 8/34 (PR8) at 2.5 Egg infectious units (EIU)/cell in a DMEM/F12 (Gibco, 21041025) growth factor-free medium. Ultraviolet light (UV)-irradiated virus that was replication-deficient (mock) was used as a control [3]. Following infection, the cells were incubated for 1 h at 37 °C, the plates were then washed twice with 2 mL PBS to remove unbound virus and supplemented with growth factor-free medium. MTECs were pretreated for 1 h with 10 μM LOC14 (5606 Tocris, Minneapolis, MN, USA), during viral infection, and 2 h post viral infection, DMSO was used as a control. All treatments were performed in growth factor-free medium [3].

### 4.4. Transfection

Cells were transfected using Lipofectamine 3000 transfection kit (L3000015 Thermo Fisher, Waltham, MA, USA) as per manufactures instructions with the following modification: 2.5 µg of plasmid per plate, optional p3000 reagent was utilized, transfections were incubated for 4 h. Human A549 adenocarcinoma alveolar basal epithelial cells were cultured as suggested by the ATCC. Cells were plated at 4 × 10^5^ cells/dish 24 h prior to transfection. Immediately prior to transfection cells were washed with 1 mL of PBS, and media replaced with 1 mL Opti-MEM (31985070 Thermo Fisher, Waltham, MA, USA). After 4 h, transfection media was aspirated and replaced with 2 mL Opti-MEM. 

### 4.5. NA Activity Assay

NA activity assay was adapted from Leang et al. [41]. The assay is based on NA enzyme activity cleaving the 2′-(4-Methylumbelliferyl)-α-D-N-acetylneuraminic acid (MUNANA) substrate (16620 Cayman, Ann Arbor, MI, USA) to release the fluorescent product 4-methylumbelliferone (4-MU). NA activity was monitored in NA activity buffer containing 33.3 mM 2-(N-morpholino) ethanesulfonic acid (MES) and 4 mM CaCl2, pH 6.5 by adding 50 µL infected cell supernatant or 5 µg of infected cell lysate to 300 µM MUNANA. The increase in fluorescence signal was monitored at 460 nm with excitation at 355 nm using a Synergy HTX plate reader (Biotek, Santa Clara, CA, USA) [3]. Five (5) μg of recombinant NA (rNA) was reacted with 2 mM DTT to reduce disulfide bonds (Figure 1B). Total of 5 μg of total protein from the cell lysates, supernatants, or BALF were used for NA assay in Figure 2E,G and Figure 4F. The total reaction volume was 100 μL. Rates were calculated using GraphPad Prism (8.4.1).

### 4.6. Dead Cell Protease Assay 

Cell death in influenza virus-infected ±LOC14-treated A549 cells was measured using the Apo Tox-Glo assay (Promega, Madison, WI, USA) dead cell protease substrate as described in Anathy et al., 2012 [30].

### 4.7. CRISPR Lines

PDIA3-deficient A549 cells were purchased from Synthego (Synthego Corporation, Menlo Park, CA, USA). Cells were screened by plating 1 cell/well in a 96 well flat bottom plate. Single cell wells were marked and monitored for growth. After approximately four doublings, cells were trypsinized and transferred to 6-well plates. Upon reaching ~70% confluency cells were trypsinized and transferred to T75 flasks. PDIA3 levels were ascertained by Western blot.

### 4.8. Mouse Model of IAV-Infection and LOC14 Treatment

All mice were females, C57BL/6NJ strain, bred, and maintained at the University of Vermont animal facility. All mice were anesthetized using isoflurane and exposed to 2000 egg infectious units (EIU) IAV (PR8) or mock (UV killed PR8) suspended in PBS via the nasopharyngeal route. Mice were administered LOC14 via the intraperitoneal injection (IP) 1 h before infection and 24 h or 48 h post infection via oropharyngeal route under anesthesia. Mice were euthanized at 72 h post infection to collect bronchoalveolar lavage fluid (BALF) and lung samples for further analysis (Figure 3A). 

### 4.9. BALF Processing

BALF was collected by lavaging lungs with 1.0 mL of sterile PBS. Cells were isolated by centrifugation, and total cell counts were determined using a Guava easyCyte HT cytometer (Millipore) and analyzed using Flowjo (version 10.4.2, Ashland, OR: Becton, Dickinson and Company, Ashland, OR, USA). Differential cell counts were obtained via cytospins using Hema3-stained (Fisher Scientific) total cells, on a minimum of 300 cells/animal. The differential cell counts were normalized to the total cell counts [3].

### 4.10. Analysis of mRNA Expression

Adapted from Chambelain et al., 2019 [3]. Right lung lobes were flash-frozen and pulverized, and total RNA was isolated using Qiazol Lysis Reagent (Qiagen, Germantown, MD, USA) and purified using the RNeasy kit (Qiagen). One microgram of RNA was reverse transcribed to cDNA using M-MHV Reverse Transcriptase (Promega, Madison, WI, USA) for quantitative assessment of gene expression using SYBR green (Bio-Rad, Hercules, CA, USA). Expression values were normalized to indicated housekeeping genes. The primers used in this study are listed in Table 1 [3]. 

### 4.11. Biotin Switch Assay

Adapted from Chamberlain et al., 2019 [3]. To block free sulfhydryls, cells were lysed in buffer containing 20 mM Tris·HCl (pH 8.0), 100 mM NaCl, 0.5% Nonidet P-40, 10% glycerol, 1% protease inhibitor cocktail (P8340 Sigma-Aldrich, St. Louis, MO, USA) (*v*/*v*), 1% phosphatase inhibitor cocktails 1 and 2 (P5726, P0044 Sigma-Aldrich, St. Louis, MO, USA) (*v*/*v*), and 1 mM N-ethylmaleimide (NEM) for 1 h at ambient temperature. Excess NEM was then removed via acetone precipitation [3]. Briefly, acetone was cooled to −20 °C. Four times the sample volume was added to protein samples. Samples were then vortexed and incubated overnight at −20 °C. Precipitated protein was pelleted by centrifugation at 14,000× *g* for 10 min. The supernatant was aspirated, and the resulting pellet was suspended in buffer containing 20 mM Tris·HCl (pH 8.0), 100 mM NaCl, 0.5% Nonidet P-40, 10% glycerol, 1% SDS, and 1% protease inhibitor cocktail (P8340 Sigma-Aldrich, St. Louis, MO, USA) (*v*/*v*), 1% phosphatase inhibitor cocktails 1 and 2 (P5726, P0044 Sigma-Aldrich, St. Louis, MO, USA) (*v*/*v*). Upon resuspension disulfides were reduced with 20 μM DTT and newly formed sulfhydryl groups were labeled with 1 mM 3-(N-maleimido-propionyl) biocytin (MPB) (M1602 Invitrogen, Waltham, MA, USA) for 1 h at ambient temperature [3]. Excess DTT and MPB were removed via acetone precipitation. The labeled lysate was precipitated using NeutrAvidin agarose resin (29200 Thermo Scientific, Waltham, MA, USA) and subsequently probed using an anti-HA antibody. As a reagent control, lysates from cells were incubated with DMSO and subjected to the same procedures [3].

### 4.12. Non-Reducing Gel Electrophoresis

Adapted from Chamberlain et al., 2019 [3]. Cell lysates were resuspended in loading buffer without the reducing agent DTT. A separate set of samples was resuspended in loading buffer with DTT and incubated at 95 °C for 5 min to reduce the disulfide bonds. The samples were resolved by SDS-PAGE and subjected to Western blot analysis as described [3].

### 4.13. Western Blot Analysis

Adapted from Chamberlain et al., 2019 [3]. Following dissection, right lung lobes were flash-frozen for protein or mRNA analysis. Lungs were pulverized and lysed in buffer containing 20 mM Tris·HCl (pH 8.0), 100 mM NaCl, 0.5% Nonidet P-40, 10% glycerol, and 1% protease inhibitor cocktail (P8340 Sigma-Aldrich, St. Louis, MO, USA) (*v*/*v*), 1% phosphatase inhibitor cocktails 1 and 2 (Sigma-Aldrich, P5726, P0044) (*v*/*v*) [3]. Proteins from cell lysates were prepared in the same buffer. Insoluble proteins were pelleted via centrifugation. Following protein quantitation of the supernatant, samples were resuspended in loading buffer with DTT and resolved by SDS-PAGE. Proteins were transferred to PVDF, and membranes were probed using a standard immunoblotting protocol. Protein quantification of supernatant was determined using DC Protein Assay (5000116 Bio-Rad, Hercules, CA, USA). Samples were resuspended in loading buffer with DTT and resolved by SDS-PAGE. The quantification of protein expression was performed by densitometry using Image Studio Lite software (LI-COR Biosciences, Lincoln, NE, USA) [3]. Antibodies used for Western blots are as follows: IAV H1 HA (11684-RP01 Sino biological, Wayne, PA, USA); IAV N1 NA (AF4858 R & D, Minneapolis, MI, USA); PDIA3 (LSB9768 LS bio, Seattle, Washington, USA); PDIA1 (ADI-SPA-890-F Enzo, New York, NY, USA); PDIA5 (NBP1-92252 Novus Bio, Centennial, CO, USA); PDIA6 (ab83456 Abcam, Waltham, MA, USA); β-actin (A5441 Sigma, St. Louis, MO, USA), and GAPDH (607902 Biolegend, San Diego, CA, USA) [3].

### 4.14. Image Processing

Digital images were acquired using an Amersham Imager 600RGB (GE). Photoshop (CC 2020; Adobe, San Jose, CA, USA) and Illustrator (CC 2020; Adobe, San Jose, CA, USA) were used to assemble the figures. Samples were run on the same gel. When required, brightness and contrast were adjusted equally in all lanes [3].

### 4.15. ELISA

Adapted from Chamberlain et al., 2019 [3]. Lung protein samples were assayed for IL-6 (DY406, R&D, Minneapolis, MN, USA), GCSF (DY414, R&D, Minneapolis, MN, USA), IL-1β (DY401, R&D, Minneapolis, MN, USA), KC (DY453, R&D, Minneapolis, MN USA), IFN-β (DY814-05 and DY8234-05, R&D, Minneapolis, MN, USA), and CCL20 (DY760, R&D, Minneapolis, MN, USA) by ELISA according to the manufacturer’s instructions.

### 4.16. Serum ALT Determination

Serum ALT levels were determined using an alanine transaminase colorimetric activity assay kit (700260 Cayman, Ann Arbor, MI, USA). Assay was run according to manufacturer’s instructions.

### 4.17. Immunoprecipitation

Adapted from Chamberlain et al., 2019 [3]. HBE cells were lysed in a buffer containing 20 mM Tris·HCl (pH 8.0), 100 mM NaCl, 0.5% Nonidet P-40, 10% glycerol, 1% protease inhibitor cocktail (P8340 Sigma-Aldrich, St. Louis, MO, USA) (*v*/*v*), 1% phosphatase inhibitor cocktails 1 and 2 (P5726, P0044 Sigma-Aldrich, St. Louis, MO, USA) (*v*/*v*). PDIA3 was precipitated using anti-PDIA3 antibody (ADI-SPA-585-F Enzo Life Sciences, Farmingdale, NY, USA) and Protein G agarose beads (15920010 Invitrogen, Waltham, MA, USA). As a control, lysates from cells were incubated in non-specific rabbit gamma globulin (011-000-002 Jackson ImmunoResearch, West Grove, PA, USA) and subjected to the same procedures. Samples were run on reducing gels [3].

### 4.18. Virus Plaque Assay

Confluent MDCK (NBL2, ATCC-CCL34) cells were trypsinized and seeded to 6-well plates at 3 × 10^6^ cells/well and incubated at 37 °C, 5% CO_2_ for overnight. The next day the cells were rinsed 2–3 times with sterile 1× PBS to remove all FBS from growth media. The cells were infected with 1.5 mL of various serial dilutions of lung homogenate and incubated at 37 °C, 5% CO_2_ for one hour. Following the virus absorption, cells were washed twice with sterile 1X PBS. Then these cells were overlaid with 3 mL of 0.5% low melting agarose (+2 ug/mL TPCK). The cells were incubated at 37 °C, 5% CO_2_ for 72 hrs. Once plaques are visible, the agarose gel was removed from each well. The agarose and cells were fixed with 2 mL 10% PFA for 1 h (at room temperature). After 1 h PFA was removed, and agarose was stained with 1 mL 1% crystal violet for 20 min (at room temperature). Excess stain was removed, and wells were washed with water. The number of visible plaques were recorded in each well, and calculated as PFU/mL = Average of plaques (in duplicate wells)/dilution × Volume.

### 4.19. Statistics

Adapted from Chamberlain et al., 2019 [3]. All mice studies were repeated once. All statistical analysis was carried out using Graph Pad Prism 8. The ROUT [42] method was used to identify outliers with a cutoff of Q = 2%, and identified outliers were removed from the subsequent statistical analysis (see Figure legends). No sample size were estimated prior to the start of the experiment. Data were pooled from two experiments and analyzed by one or two-way analysis of variance (ANOVA) where appropriate, and a two-stage linear step-up procedure of Benjamini, Krieger, and Yekutieli [43] (FDR, Q = 5% or 0.05) test to adjust for multiple comparisons. Adjusted *p* values (“q” values) of <0.05 were regarded as discovery in FDR. Parametric student’s *t*-test were used where appropriate. *p* values < 0.05 were regarded as statistically significant in Mann Whitney and parametric student’s *t*-test. Data for all the results were expressed as ±SEM.

## Figures and Tables

**Figure 1 ijms-23-01078-f001:**
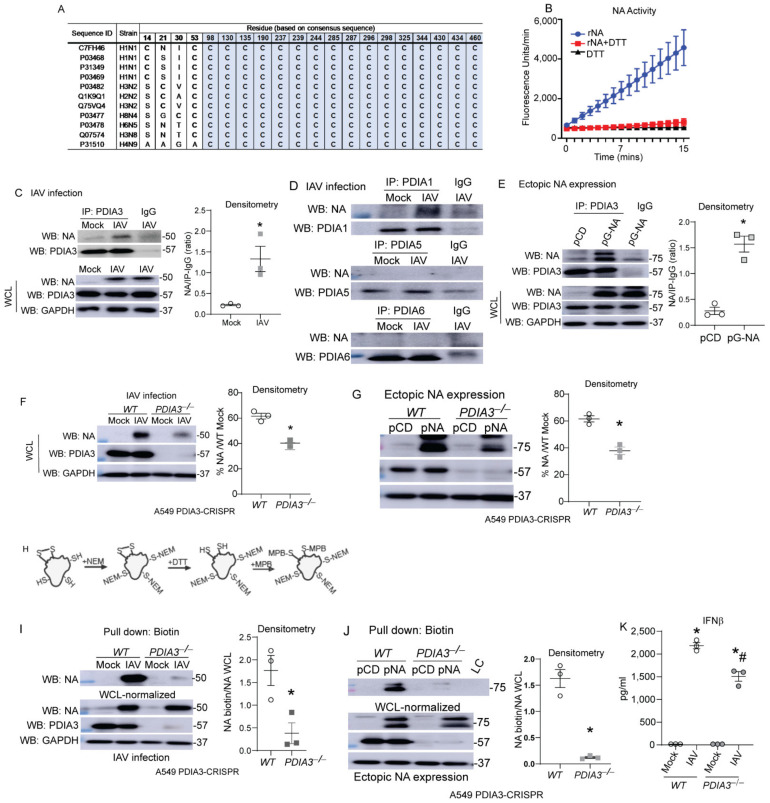
PDIA3 interaction with influenza NA required for disulfides of NA. (**A**) Multiple alignments of conserved cysteine residues that are involved in disulfide bonds (grey) of NA. (**B**) Recombinant NA (rNA-5 μg) activity assay (±reducing agent-DTT-1 mM). (**C**,**D**) NA interaction with PDIA3/PDIA1/A5/A6 assessed by respective PDIA immunoprecipitation in mock or influenza-infected cells. (**E**) pCDNA3 or pNA-transfected A549 cells, NA interaction assessed by PDIA3 immunoprecipitation. (**F**,**G**) Expression of NA in PDIA3 ablated (*PDIA3*^−/−^) or WT A549 cells infected (left) or transfected with plasmids (right) with densitometry. (**H**) Schematic depicting disulfide labeling assay. (**I**,**J**) Western blots for NA labeling of disulfides in samples normalized with higher total protein in *PDIA3*^−/−^ A549 cells infected (left) or transfected with plasmids (right) with densitometry of disulfide labeled proteins. (**K**) IFNβ measurement by ELISA. * *p* < 0.05 compared to mock or WT groups by *t*-test. # *p* < 0.05 compared WT + IAV group. Data are expressed as the standard error of the mean (±SEM).

**Figure 2 ijms-23-01078-f002:**
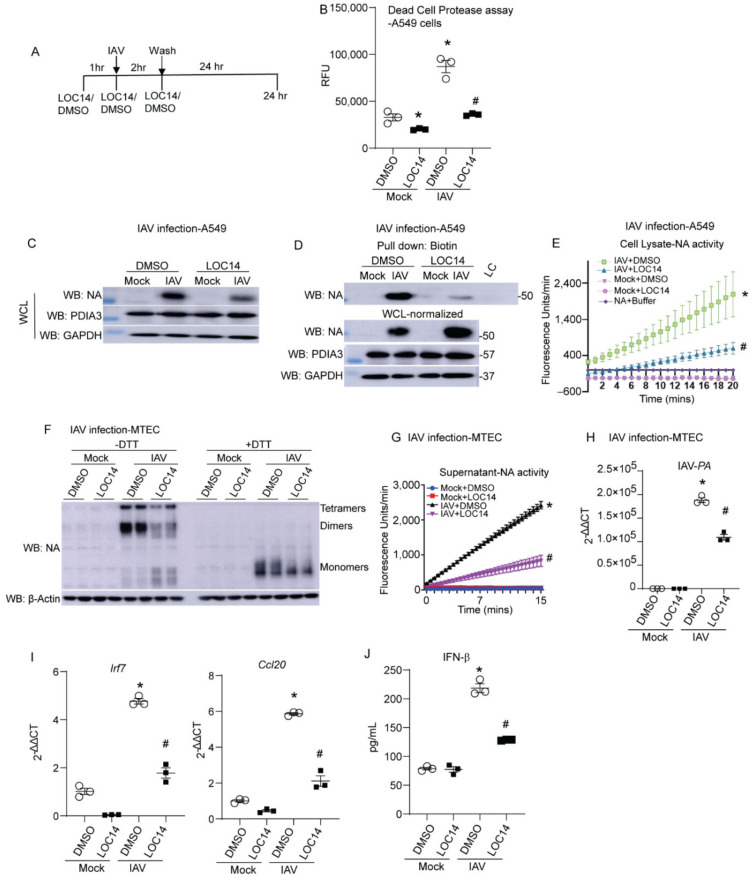
PDI inhibitor LOC14 decreases disulfides in NA, the activity of NA, viral burden, and pro-inflammatory response in primary lung epithelial cells. (**A**) Schematic representing the time points of IAV infection and LOC14/DMSO treatment and harvest in cells. (**B**) Dead cell protease assay (RFU—relative fluorescence units). (**C**) Western blot analysis of influenza NA, PDIA3 in H1N1 PR8 or mock virus-infected A549s treated with vehicle control (DMSO) or LOC14. (**D**) Western blots of NA labeled for disulfides in samples spiked with 1.5× higher total protein in cells treated with LOC14. (**E**) NA activity from the cell lysates infected with H1N1 PR8 or mock virus and treated with vehicle control (DMSO) or LOC14. (**F**) Oligomerization of NA in IAV or mock virus-infected MTECs treated with vehicle control (DMSO) or LOC14, Western blot ±DTT. (**G**) Influenza-NA activity measurement from cell supernatants of MTECs infected with IAV or mock virus and treated with vehicle control (DMSO) or LOC14. (**H**) Influenza-PA mRNA measurement from cell lysates of MTECs infected with H1N1 PR8 or mock virus and treated with vehicle control (DMSO) or LOC14. (**I**) Measurement of Interferon regulatory factor 7 (Irf7) or Ccl20 mRNA. (**J**) Measurement of IFNβ by ELISA. * *p* < 0.05 compared to mock + DMSO group, # *p* < 0.05 compared to IAV + DMSO group, by ANOVA. Data are expressed as the standard error of the mean (±SEM).

**Figure 3 ijms-23-01078-f003:**
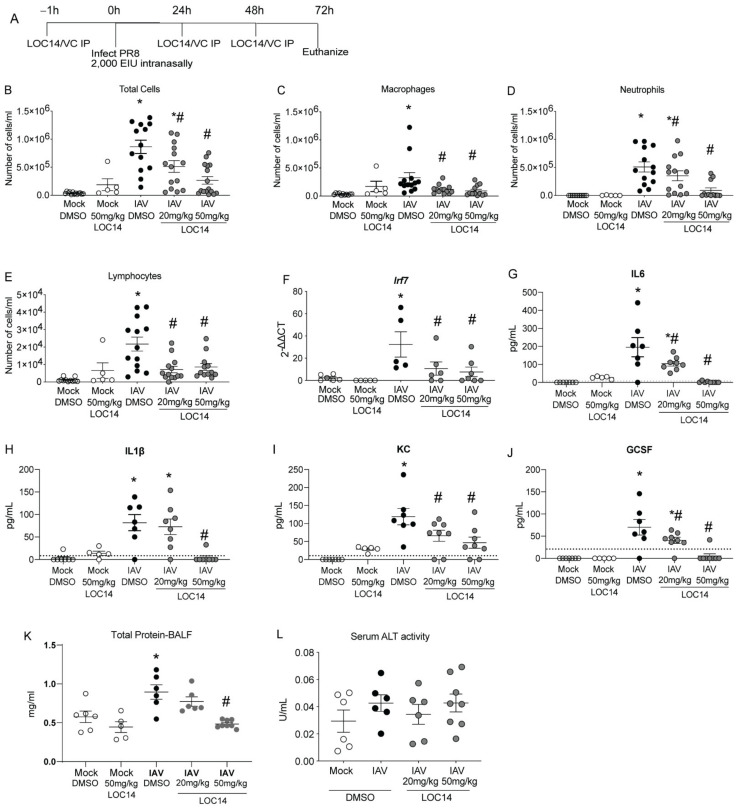
LOC14 treatment decreases influenza-induced inflammatory response in mice. (**A**) Schematic representing the time points of IAV infection and LOC14 treatment and euthanasia of mice (IP-intraperitoneal injection). (**B**–**J**) Analysis of inflammatory cells by flow cytometry, mRNA for Irf7 in whole lung lysate by RT-qPCR, and ELISA for inflammatory cytokines and chemokines from BAL fluid of infected mice. (**K**,**L**) Analysis of total protein in BALF and serum ALT in influenza or mock-infected mice treated with LOC14. * *p* < 0.05 compared to mock groups, # *p* < 0.05 compared to IAV-DMSO group by two-way ANOVA. Data are expressed as the standard error of the mean (±SEM).

**Figure 4 ijms-23-01078-f004:**
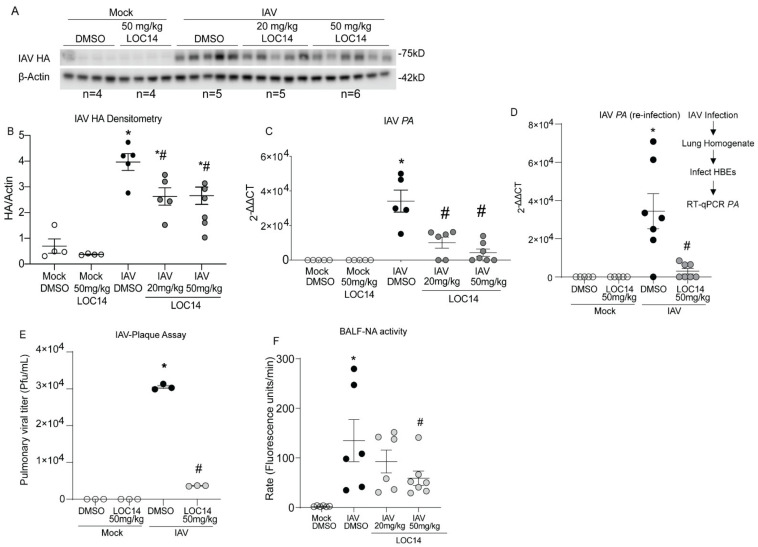
LOC14 treatment decreases the influenza burden in mice. (**A**,**B**). Western blot analysis of IAV HA and densitometry of HA normalized to actin. (**C**,**D**) Analysis mRNA for influenza PA in whole lung lysate by RT-qPCR and analysis mRNA by RT-qPCR for influenza PA from the lung homogenate re-infected into HBE. (**E**) IAV plaque formation assay. (**F**) Analysis of NA activity in the BALF. * *p* < 0.05 compared to mock groups, # *p* < 0.05 compared to IAV-DMSO group by two-way ANOVA. Data are expressed as the standard error of the mean (±SEM).

**Table 1 ijms-23-01078-t001:** List of qPCR primers for RT-qPCR analysis in this study.

Primer Name	Primer Sequence (5′-3′)
Polymerase acidic–FW	CGGTCCAAATTCCTGCTGA
Polymerase acidic–REV	CATTTGGGTTCCTTCCATCC
mPP1B–FW	TTTTCATCTGCACTGCCAAG
mPP1B–REV	TGCAGTTGTCCACAGTCAGC
mRP2–FW	TTGCCAGCAATTCGTGTGA
mRP2–REV	CCAGTTGACCTCTTCTGACA
mGAPDH–FW	AGGTCGGTGTGAACGGATTTG
mGAPDH–REV	TGTAGACCATGTAGTTGACCTCA
mIRF7–FW	GAAGACCCTGATCCTGGTGA
mIRF7–REV	CCAGGTCCATGAGGAAGTGT

## Data Availability

All data supporting the findings of this study are available within this manuscript.

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
