# Peer review of "Protein Disulfide Isomerase A3 Regulates Influenza Neuraminidase Activity and Influenza Burden in the Lung"

_ijms, 2022, doi:10.3390/ijms23031078_

Round 1

Reviewer 1 Report

In this manuscript, the authors investigated the role of protein disulfide isomerase A3 (PDIA3) in influenza virus (IAV) neuraminidase (NA) activity. The authors have shown that IAV-NA interacts with PDIA3 which is critical for NA function and inhibition of PDIA3 reduced viral burden and inflammatory responses in both in-vitro and in-vivo models. This is a well-written manuscript and the experiments are well-designed to address the research questions. However, the study lacks mechanistic insights on how PDIA3 inhibition attenuates viral burden and ameliorates inflammation.

  • It is not clear why only the airway epithelial cells were used for experiments. Please mention the rationale of such an approach.
  • Is the protective effect of LOC14 driven by inhibition of viral propagation only or associated with upregulation of antiviral immunity? Further mechanistic data will add to the novelty and significance of the study.
  • Does LOC14 lower lung pathologies?
  • Since LOC14 alters functions of both HA and NA. It’s not possible to conclude the observed anti-viral effect is entirely PDIA3-NA dependent.
  • PDIA3 is vital for the homeostatic functioning of cellular proteins. Does inhibition of PDIA3 affect a broad array of cellular proteins? Does LOC14 affect cell survival?
  • PDIA3 inhibition has a broader effect on the stabilization and functioning of cytokines and chemokines. Hence, the effect of LOC14 cannot be ascertained to NA or HA only. Is the attenuated inflammatory response by LOC14 due to the lack of functional proteins? In such a case, the title of the manuscript is misleading.
  • A comparison between NA inhibitors and LOC14 should be performed to examine how much of the observed effect of LOC14 is NA dependent. 
  • References 33 to 36 haven’t been cited in the text.
  • References 38 and 39 do not support the text cited.

Reviewer 2 Report

This is a very interesting and well-conducted article using both in vitro and in vivo models of diseases with genetic deletion. I do not have any major comment to improve and would congratulate authors for theri work.

Author Response

Thank you for your kind comments on our MS.

Reviewer 3 Report

This is a very interesting study, well-written and with novel and robust findings.

I have two comments:

  • Please expand discussion section, with more extensive comparisons with previous literature and better highlighting the stong points of the study
  • Did authors estimate sample size? Please specify it in statistics section.

Round 2

Reviewer 1 Report

The authors have satisfactorily addressed the comments. Congratulation on the great work and looking forward to reading the next mechanistic paper mentioned in the author's response.